# Psychological resilience mediates the association between sleep quality and anxiety symptoms: A repeated measures study in college students

**Huangjie Cai** [1]*, **Jianhui Guo**[2], **Jungu Zhou**[2], **Yingqian Lai**[3]

**1** Mental Health Guidance Center, Fujian Medical University, Fuzhou, China, **2** Department of Epidemiology and Health Statistics, School of Public Health, Fujian Medical University, Fuzhou, China, **3** School of Public Health, Fujian Medical University, Fuzhou, China

* 165475322@qq.com

## Abstract

### Objective

To explore the association between sleep quality and anxiety symptoms, and the mediation effect of psychological resilience on this association by a repeated measures study.

### Methods

In this study, 127 college students were randomly recruited and their sleep quality and psychological status were repeatedly collected using the Pittsburgh Sleep Quality Index (PSQI) scale, Connor-Davidson resilience scale (CD-RISC), and 7-items generalized anxiety disorder scale (GAD-7). Linear mixed-effects models were used to explore the association between sleep quality and anxiety symptoms, and a mediated effects analysis was used to explore the role played by psychological resilience in this association.

### Results

This study found a positive association between sleep quality and anxiety symptoms ($\beta = 0.40$, 95% confidence interval: 0.27, 0.52). Psychological resilience scores and its various dimensions play a significant mediating role in this association.

### Conclusions

Although the role of sleep quality in anxiety disorders is not fully understood, this study highlights the importance of improving sleep quality while enhancing psychological resilience to prevent the onset of anxiety symptoms in college students.

**Data Availability Statement:** All relevant data are included in the manuscript files.

**Funding:** The author(s) received no specific funding for this work.

**Competing interests:** The authors have declared that no competing interests exist.

# 1 Introduction

Sleep plays an important role in maintaining normal physiological function [1], and changes in sleep patterns can come with many health problems [2–4]. Increased caffeine intake, stress and irregular sleep-wake patterns lead to widespread sleep disorders and sleep deprivation in the college population [5]. Previous studies have found that one third of the Chinese college students has problems with sleep quality [6]. In addition, mental health problems (such as anxiety) are widespread in the college population, especially during the novel coronavirus epidemic [7, 8]. Previous studies have found a strong association between the quality of sleep of college students and the onset of their anxiety symptoms [9, 10]. However, most of the studies are based on cross-sectional studies. Therefore, there is a need for prospective studies to explore the association between sleep quality and anxiety.

Psychological resilience refers to the ability of an individual to adapt psychologically or behaviorally to reduce the level of stress and minimize the damage to the organism when exposed to external stress [11]. Several studies have examined the mediating role played by psychological resilience in the association between external influences and mental health [12, 13]. Current research has found that psychological resilience is negatively associated with the risk of anxiety [14]. Healthy lifestyle habits help to improve the psychological resilience of individuals [15, 16]. In contrast, people with poor sleep quality have poor psychological stress capacity, which increases the likelihood of anxiety.

Therefore, the present study used a repeated measures approach, using the 7-items generalized anxiety disorder scale, the Pittsburgh sleep quality index scale, and the Connor-Davidson resilience scale to repeatedly collect indicators of participants' mental health status and sleep status to explore the association of sleep quality with anxiety symptoms and the mediating role of psychological resilience in this association in a college student population.

# 2 Method

## 2.1 Study participants

In this study, a convenience sampling method was used to collect 127 college students in school. The participants study participated in an on-site electronic questionnaire for two repeated measures. The two survey periods of this study were April 5, 2022-April 12, 2022 and May 27, 2022-June 3, 2022, respectively. During this period, the occurrence of the epidemic and the school being under closed management may have affected the sleep status and mental health of some students. This study complied with the Declaration of Helsinki and approved by the Ethics Committee of Fujian medical university.

## 2.2 Study measures

**7-items generalized anxiety disorder scale (GAD-7).** The seven items of the GAD-7 describe the most important diagnostic criteria for GAD according to DSM-IV-TR [17]. The GAD-7 assess the frequency of generalized anxiety-related symptoms in the last 2 weeks. The scale is based on a four-point Likert scale, ranging from "not at all" to "almost every day", with scores ranging from 0 to 3, and total scores ranging from 0 to 21, with higher total scores indicating more severe anxiety symptoms [18]. The Cronbach's alpha coefficient of this scale was 0.92, which has good reliability and validity [19].

**Connor-Davidson resilience scale (CD-RISC).** The CD-RISC consists of 4 dimensions and 25 items to assess the psychological resilience of college students [20]. The Cronbach's alpha coefficient of the Chinese version of the CD-RISC is 0.91, which has good convergent and discriminant validity [21]. Each item is scored on a five-point Likert scale, ranging from "never"

to "almost always" with score assigned from 1 to 5. The total score ranged from 25 to 125 and the higher the score, the higher the level of psychological resilience of the individual. The 25 items of the scale can be divided into 3 dimensions, namely optimism, resilience and strength [21].

**Pittsburgh sleep quality index (PSQI) scale.** The PSQI scale was used to assess sleep quality in the last 1 month [22]. The Cronbach's alpha coefficient of the scale was 0.92, with good reliability [23]. The PSQI scale consists of 7 items, and each item is scored inversely, from "very good" to "poor", using the Likert four-point scale. The higher the score, the worse the sleep quality.

## 2.3 Statistical methods

In this study, the measurement data that did not conform to normal distribution were expressed as median (minimum, maximum) and the count data were expressed as frequency (percentage). Correlation analysis was used to test the degree of association between the GAD score, CD-RISC score and PSQI. We used the paired Wilcoxon rank sum test to test whether the GAD score, CD-RISC score, and PSQI differed between the two surveys. Linear mixed-effects models were used to examine the relationship between sleep quality and psychological resilience and generalized anxiety symptoms. We used stratified analysis to test the stability of the association of sleep quality and psychological resilience with generalized anxiety. According to previous studies [24], mediating effects analysis was used to test whether psychological resilience played a mediating effect in the association between sleep quality and generalized anxiety by linear mixed-effects model. The mediation effect was calculated using the Sobel test [25]. All analyses were performed using the R language. $P < 0.05$ was considered statistically significant.

## 3 Results

### 3.1 Study population

A total of 127 college students were recruited in this study, including 66 (52%) males, 95 (74.8%) younger than 20 years old, 83 (65.4%) with Body Mass Index (BMI) at normal levels (18.5 kg/m$^2$-23.9 kg/m$^2$), and 68 (53.5%) who exercised 1 to 3 times/week during the survey period (Table 1). The results of the first survey and the second survey showed significant correlations among the PSQI, GAD score, and CD-RISC score. The CD-RISC score and its dimensions were negatively correlated with the PSQI or CD-RISC score and GAD score, while the GAD score and the PSQI were positively correlated (Fig 1).

### 3.2 Consistency check of PSQI, CD-RISC, and GAD scale

In this study, the PSQI scale, psychological resilience scale and generalized anxiety disorder scale of the two pre- and post-surveys were tested for consistency, and it was found that the Cronbach's alpha coefficient of all three scales was greater than 0.6, suggesting that the consistency of all three scales was good (as shown in Table 2).

### 3.3 Comparison of PSQI and CD-RISC scores between the two surveys

The results of the two surveys showed that among college students, the median GAD-7 scores were 10 and 11, respectively, and the difference between them was statistically significant ($Z$ = -3.042, $P$ = 0.002). In both surveys, the median sleep time scores were 1 and 1 ($Z$ = -2.844, $P$ = 0.025), respectively, with statistically significant differences between the two. In addition, no statistically significant differences in psychological resilience and sleep quality among college students before and after the lifting of closed management in colleges and universities have been found in both surveys (Table 3).

**Table 1. Description characteristics of participants in this study.**

| Variables | n(%) |
|---|---|
| Sex | |
| Male | 66(52.0) |
| Female | 61(48.0) |
| Age (years) | |
| $\leq$20 | 95(74.8) |
| $\geq$21 | 32(25.2) |
| BMI (kg/m$^2$) | |
| <18.5 | 22(17.3) |
| 18.5–23.9 | 83(65.4) |
| 24–27.9 | 12(9.4) |
| $\geq$28 | 10(7.9) |
| Physical exercise (times/week) | |
| <1 | 32(25.2) |
| 1–3 | 68(53.5) |
| 4–5 | 11(8.7) |
| >5 | 16(12.6) |

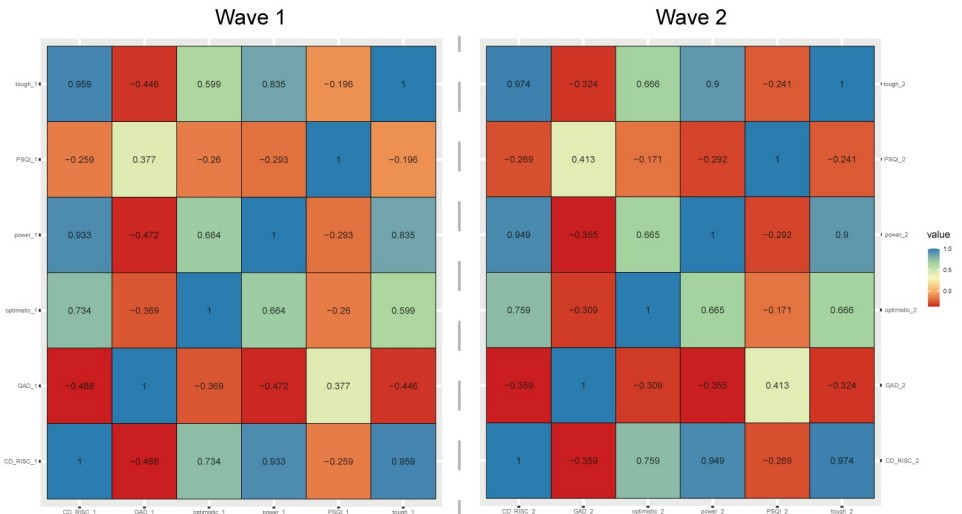

**Fig 1. Correlation of PSQI, CD-RISC score, and GAD score.**

## 3.4 Association of sleep quality and psychological resilience with anxiety symptoms

After adjusted potential confounders, we found that PIQI score was positive associated with anxiety symptoms ($\beta$ = 0.40, 95% *CI*: 0.27, 0.52) and CD-RISC score was negative associated

**Table 2. Consistency check of PSQI scale, CD-RISC, and GAD-7.**

| Variables | Wave 1 | Wave 2 |
|---|---|---|
| CD-RISC | 0.952 | 0.952 |
| GAD-7 | 0.903 | 0.903 |
| PSQI scale | 0.638 | 0.732 |

**Table 3. Comparison of PSQI and CD-RISC scores between the two surveys.**

| Variables | Wave 1 (n = 127) | | Wave 2 (n = 127) | | Z | P |
|---|---|---|---|---|---|---|
| | M | min, max | M | min, max | | |
| GAD score | 10 | 7, 24 | 11 | 7, 21 | **-3.042** | **0.002** |
| CD-RISC score | 84 | 50, 125 | 84 | 25, 122 | -0.063 | 0.950 |
| Tough | 42 | 23, 65 | 42 | 13, 65 | -0.716 | 0.474 |
| Power | 29 | 16, 40 | 28 | 8, 40 | -1.241 | 0.215 |
| Optimistic | 13 | 6, 20 | 13 | 4, 20 | -0.189 | 0.850 |
| PSQI | 5 | 0, 15 | 5 | 0, 18 | -1.374 | 0.170 |
| Sleep quality | 1 | 0, 3 | 1 | 0, 3 | -1.098 | 0.272 |
| Sleeping time | 1 | 0, 3 | 1 | 0, 1 | **-2.844** | **0.025** |
| Sleeping time | 0 | 0, 3 | 0 | 0, 2 | -1.040 | 0.298 |
| Sleep efficiency | 0 | 0, 3 | 0 | 0, 3 | -1.715 | 0.086 |
| Sleep disorders | 1 | 0, 2 | 1 | 0, 3 | <0.001 | 1.000 |
| Hypnotic drugs | 0 | 0, 3 | 0 | 0, 3 | -1.310 | 0.190 |
| Daytime dysfunction | 1 | 0, 3 | 1 | 0, 3 | -0.825 | 0.409 |

Abbreviation: M, Median; min, Minimum; max, Maximum.

**Table 4. Association of sleep quality and psychological resilience with generalized anxiety disorder.**

| Variables | β (95% confidence interval) | |
|---|---|---|
| | Model 1 | Model 2 |
| PSQI score | 0.40(0.28, 0.52) | 0.40 (0.27, 0.52) |
| CD-RISC score | -0.08(-0.11, -0.05) | -0.08(-0.11, -0.05) |

Model 1 was the unadjusted model.

Model 2 adjusted for sex, age, BMI, physical exercise.

with anxiety symptoms ($\beta$ = -0.08, 95% *CI*: -0.11, -0.05). In order to verify the robustness of these associations, stratified analysis by sex and age showed that the association between GAD score or CD-RISC score and anxiety symptoms was stable in all subgroups (as shown in Tables 4 and 5).

## 3.5 Mediation of psychological resilience

The mediation analysis showed that psychological resilience played a significant mediated effect on the association between sleep quality and anxiety symptoms (P<0.05). Further exploring the mediating effect played by the various dimensions of psychological resilience, we

**Table 5. Association of sleep quality and psychological resilience with generalized anxiety disorder, stratified by sex and age.**

| Variables | β (95% confidence interval) | | | |
|---|---|---|---|---|
| | Sex | | Age (years) | |
| | Male | Female | ≤20 | ≥21 |
| PSQI score | 0.44(0.27, 0.61) | 0.35(0.18, 0.52) | 0.32(0.17, 0.47) | 0.60(0.38, 0.82) |
| CD-RISC score | -0.07(-0.11, -0.02) | -0.10(-0.14, -0.06) | -0.08(-0.11, -0.04) | -0.08(-0.14, -0.02) |

β (95% confidence interval) was calculated by Linear mixed-effects model after adjusted for sex, age, BMI, and physical exercise.

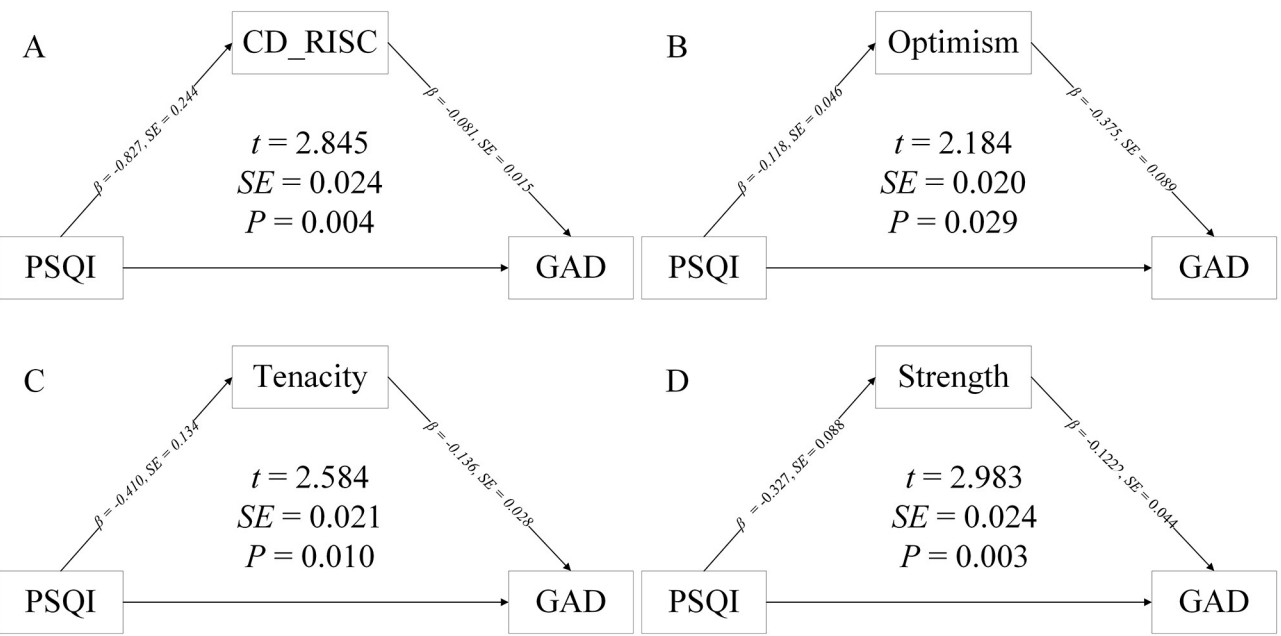

**Fig 2. Mediation of association between sleep quality and generalized anxiety disorder by psychological resilience.** Optimistic, tough, and power presenting the three dimensions of CD-RISC. β and standard error (SE) was calculated by Linear mixed-effects model after adjusted for sex, age, BMI, physical exercise.

found that power dimension, tough dimension and optimistic dimension played a significant mediated effect on these association (P<0.05). (as shown in Fig 2).

## 4 Discussion

In this study, we recruited a total of 127 university students to repeat the survey on their psychological status and sleep quality. We have found that poor sleep quality can lead to the development of anxiety symptoms. Psychological resilience plays a mediating role in the association between sleep quality and the onset of anxiety symptoms.

First, our study found significant associations between sleep quality, anxiety symptoms and psychological resilience and its various dimensions. Several studies were consistent with our results. For example, a meta-analysis of randomized controlled trials found that improved sleep quality could help reduce the risk of anxiety [26]. Another cross-sectional study also found a significant association between PSQI and GAD scores [27]. A potential mechanism by which poor sleep quality affects mental health is that poor sleep quality may affect the neuroendocrine system, particularly leading to activation of the hypothalamic-pituitary-adrenal axis and elevated glucocorticoids, thereby affecting mental health [28]. In addition, previous studies also have found a negative correlation between psychological resilience and its various dimensions and anxiety [29, 30]. The possible reason for this is that psychological resilience contributes to positive emotional recovery [16]. Further, some other studies have found an association between sleep quality and psychological resilience [15, 31]. The possible reason for this is that poor sleep quality leads to poor psychological stress capacity.

Second, this study identified the mediating role of psychological resilience. Psychological resilience, as an important psychological trait closely related to an individual's physical and mental health, is an individual's ability to maintain or return to psychological health and functioning in the face of adversity and is influenced by a variety of factors, including innate

factors, as well as acquired factors such as education and environment [11]. To our knowledge, there are relatively few studies that have examined the mediating role of psychological resilience in sleep quality and anxiety symptoms. However, a previous study found a mediating role for psychological resilience in the association between sleep and depression in a college student population [32]. In addition to this, several studies have found the mediating role played by psychological resilience in the association between external influences and mental health [12, 13]. Furthermore, the present study found that dimensions of psychological resilience also played a significant mediating role in this association. The current study found that the resilience dimension refers to consciously coping with setbacks when in distress and is considered a reliable buffer against stressful events and illness [21], whereas the optimism dimension is expressed as a generally positive attitude and belief about adverse situations and risky events, and the strength dimension emphasizes a return to a previous state of life [33], and when faced with the adverse effects of poor sleep quality, individuals with high psychological resilience will take measures with a positive mindset to cope with the adverse effects of poor sleep quality in order to return to their previous state.

## 5 Strengths and limitations

To our knowledge, this study is one of the few to examine the association between sleep quality and anxiety symptoms using a repeated measures design, which may provide an opportunity to explore the triangular causal relationship between sleep quality, anxiety symptoms, and psychological resilience. Mediating effect analysis suggested a mediating role for psychological flexibility in sleep quality and anxiety symptoms, laying the foundation for subsequent research. However, this study also had some limitations. First, our sample size which met the statistical requirements is a little small and further validation of our obtained association with a larger sample is needed. Second, the sleep quality, anxiety symptoms, and psychological resilience scores involved in this study were obtained based on scales that could be replaced by more objective indicators.

## 6 Conclusions

In this repeated measure study, poorer sleep quality was associated with more severe anxiety symptoms. Psychological resilience partially mediates the association between sleep quality and anxiety symptoms. This study highlights the importance of improving sleep quality while enhancing psychological resilience to prevent the onset of anxiety symptoms in college students.

## Supporting information

**S1 Data.**
(XLSX)

## Author Contributions

**Conceptualization:** Huangjie Cai, Yingqian Lai.

**Data curation:** Huangjie Cai, Jungu Zhou.

**Formal analysis:** Jianhui Guo.

**Investigation:** Jungu Zhou.

**Methodology:** Huangjie Cai, Yingqian Lai.

**Writing – original draft:** Huangjie Cai, Jianhui Guo, Jungu Zhou.

**Writing – review & editing:** Huangjie Cai, Yingqian Lai.

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
