## [Decision Letter · Decision Letter 0]

6 Mar 2023

PONE-D-23-03587Psychological resilience mediates the association between sleep quality and anxiety symptoms: Base on a longitudinal studyPLOS ONE

Dear Dr. Cai,

Thank you for submitting your manuscript to PLOS ONE. After careful consideration, we feel that it has merit but does not fully meet PLOS ONE’s publication criteria as it currently stands. Therefore, we invite you to submit a revised version of the manuscript that addresses the points raised during the review process.

We look forward to receiving your revised manuscript.

Kind regards,

Mehmet Emin Demirkol

Academic Editor

PLOS ONE

Reviewers' comments:

Reviewer's Responses to Questions

**Comments to the Author**

1. Is the manuscript technically sound, and do the data support the conclusions?

Reviewer #1: Partly

Reviewer #2: Partly

2. Has the statistical analysis been performed appropriately and rigorously? 

Reviewer #1: Yes

Reviewer #2: N/A

3. Have the authors made all data underlying the findings in their manuscript fully available?

Reviewer #1: No

Reviewer #2: Yes

4. Is the manuscript presented in an intelligible fashion and written in standard English?

Reviewer #1: Yes

Reviewer #2: No

5. Review Comments to the Author

Reviewer #1: ‘This study complied with the Declaration of Helsinki and approved by the 76 Ethics Committee of Fujian medical university.’

• Reference must be added

• Line 154 shows a letter error. (PIQI)

• They stated that the scales were applied repeatedly. It has been reported that there is a significant difference between the two measurements in GAD scores. It is recommended that this measurement difference be discussed.

‘Another cross-sectional study also found a significant association between PSQI and GAD scores’

• The direction of the relationship in the statement should be explained.

There are publications in the literature suggesting that sleep quality has a mediating role in anxiety symptoms. It should be explained why the mediation model was planned on the path from stressful life events to anxiety symptoms. The chronological emergence of the factors included in the mediation model should be mentioned. Chronologically, using the previously revealed factor as a mediator does not seem to be very compatible with the model.

'A potential mechanism by which poor sleep quality affects mental health is that poor sleep quality may affect the neuroendocrine system, particularly leading to activation of the hypothalamic-pituitary-adrenal axis and elevated glucocorticoids, thereby affecting mental health.'

• This situation is not included in the study data or the introduction part of the article. It is recommended to explain the result for another possible reason or to make a connection with the subject in the introduction.

‘Studies also 200 have found a negative correlation between psychological resilience and its various dimensions and anxiety. The possible reason for this is that psychological resilience contributes to positive emotional recovery’

• This statement should be detailed. Positive emotional recovery is often used for depressive processes. Its connection with anxiety should be detailed.

‘Further, some other studies have found an association between sleep quality and psychological resilience. The possible reason for this is that poor sleep quality leads to poor psychological stress capacity.’

• In this study, in which cross-sectional data were analyzed, it is not a direct result that low sleep quality causes psychological stress. However, it can be said that they are related. The explanation should be made with the support of the literature.

‘This study highlights the importance of improving sleep quality while enhancing psychological resilience to prevent the onset of anxiety symptoms in college students.’

• From what data is the conclusion about the onset of anxiety symptoms derived? The study's statistical analysis is cross-sectional and it doesn't seem easy to predict the cause-effect relationship.

,

'Therefore, there is a need for prospective studies to explore the association between sleep quality and anxiety.'

• In the study presented in the longitudinal design, but there is only one measurement statistical analysis. It is recommended to elucidate the contribution of the longitudinal pattern to the study.

Reviewer #2: Thanks for the opportunity to review this paper. The researchers investigated the relationship between sleep quality and anxiety symptoms in college students and the mediating role of psychological resilience in this relationship. The researchers used the Pittsburgh Sleep Quality Index, Connor-Davidson Resilience, and the 7-item generalized anxiety disorder scale. My opinions about the article are given below.

- There are some grammatical errors in the article, and I suggest language editing by the native speaker.

- In the last paragraph of the introduction, researchers should include the purpose, hypotheses, and expected possible contributions of the study, not the method and materials of the study.

- Researchers should elaborate the introduction based on the hypotheses of the study. Past studies should be utilized, and particularly why college students were evaluated.

- How did the researchers determine the sample size? Did they conduct a power analysis?

- Study participants should be written in more detail. How were the participants reached? Was informed consent obtained? Were any participants excluded from the study? What were the inclusion criteria?

- Why did the researchers assess generalized anxiety symptoms using a DSM-IV-TR scale instead of DSM-5?

- The CD-RISC scale description should be reviewed. In the first sentence, it is stated that it consists of 4 dimensions. As far as I know, it has three dimensions: tenacity, strength, and optimism. The dimension names are also written differently, so the authors should check this area. A similar situation is also valid for figure-2. In Figure-2, CD-RISC subdimensions are defined with different names.

- Scale names and abbreviations should be expressed in the same terms throughout the manuscript.

- 'it was found that the Cronbach's alpha coefficient of all three scales was greater than 0.6, suggesting that the consistency of all three scales was good.'

It is appropriate to add references for the statement (Lines 133-134).

- Table-3 should be renamed.

- What is the difference between the two separate 'sleeping time' headings in Table-3?

- Since a diagnosis of mental illness cannot be made using only the scale, statements such as generalized anxiety disorder, etc., should be avoided, as in Tables 4 and 5. I suggest interpreting the results as anxiety symptoms. Therefore, the authors should rename table-4-5 and figure-2.

- Models 1 and 2 in Table-4 should be explained in detail in the text.

- Mediation analysis results should be written in detail. Such as whether CD-RISC and its subheadings have a partial or full mediator role. Since anxiety symptoms may also affect sleep quality, why did the researchers not also investigate the relationship between GAD-7 (X) and PSQI (Y) (X→M→Y)? There may be a bidirectional relationship between GAD-7 and PSQI scores.

- In the first paragraph of the discussion (lines 186-190), the authors made inferences about the onset of anxiety symptoms. However, I am not sure that the development of anxiety symptoms can be inferred from the study data. I think the interpretation of increase or decrease would be more appropriate.

- The researchers administered the scale to the participants at approximately 1-2 months intervals. During this time, anxiety symptoms and sleep quality may be affected by many variables (students may become infected, hospitalized and receive treatment, etc.), including the epidemic period. Therefore, I do not consider it sufficient to associate the study results only with anxiety and sleep.

- References should be organized according to the journal rules.

- I think that it is not appropriate to publish the study as it is, but it is essential to re-evaluate it after the researchers develop it.

Best regards

6. PLOS authors have the option to publish the peer review history of their article (what does this mean?). If published, this will include your full peer review and any attached files.

Reviewer #1: No

Reviewer #2: No

---

## [Author Response · Author response to Decision Letter 0]

21 Apr 2023

April 17th, 2023

Mehmet Emin Demirkol, MD, PhD

Editor

PLOS ONE

Dear Dr. Demirkol,

Please find enclosed our revised manuscript entitled “Psychological resilience mediates the association between sleep quality and anxiety symptoms: Base on a longitudinal study (PONE-D-23-03587).” We appreciate the issues raised by the reviewers and the editors and are grateful for the opportunity to respond. Below we address the queries raised during the review process.

We believe that the editor and reviewer comments were very helpful and addressing them has strengthened the manuscript, and we hope you agree. If there are any further issues or concerns that we can address, please do not hesitate to contact us. 

Sincerely,

Dr Huangjie Cai,

1 Xuefu North Road, Fuzhou University New District

Email: 165475322@qq.com

Tel: +86 13774515133

Below are the original review comments, and our responses are in blue text under each one.

-Reviewer#1

1. ‘This study complied with the Declaration of Helsinki and approved by the 76 Ethics Committee of Fujian medical university.’

• Reference must be added

Thank you for your comments and suggestions. However, our ethics have not yet been published and we can only provide ethics numbers. We have supplemented our ethics number in Methods.

This study complied with the Declaration of Helsinki and was approved by the Ethics Committee of Fujian Medical University (Ethics NO.123 of 2021).

2. Line 154 shows a letter error. (PIQI)

We sincerely apologize for our careless mistakes. The misspelling is revised in our resubmitted. Thanks for your correction. The details are shown as followed:

‘After adjusted potential confounders, we found that PSQI score was positive associated with anxiety symptoms’

3. They stated that the scales were applied repeatedly. It has been reported that there is a significant difference between the two measurements in GAD scores. It is recommended that this measurement difference be discussed.

Thank you for your comments and suggestions. It was a population-based longitudinal study that assessed university students during the initial and post-closure periods when the campus adopted close management due to the epidemic. To reduce pathological factors affecting anxiety symptom, infected students were excluded from the study. According to previous studies, close management could have an impact on anxiety status which may be the reason for the significant difference between the two measurements in GAD scores. The discussion about that was added and the details are shown as followed:

The campus closure policy is one of the public health responses emphasizing social distancing, implemented in many cities in China during the outbreak. Previous Findings suggested that there were negative mental health correlates of social distancing, including anxiety (PMID: 32861098). That may be the reason for the significant difference between the two measurements in GAD scores in this study.

4. ‘Another cross-sectional study also found a significant association between PSQI and GAD scores’

• The direction of the relationship in the statement should be explained. There are publications in the literature suggesting that sleep quality has a mediating role in anxiety symptoms. It should be explained why the mediation model was planned on the path from stressful life events to anxiety symptoms. The chronological emergence of the factors included in the mediation model should be mentioned. Chronologically, using the previously revealed factor as a mediator does not seem to be very compatible with the model.

Thank you for your comments. In this study, what we mainly explored is that psychological resilience played the mediating role in this association between sleep quality and anxiety symptoms. Previous research has found that psychological resilience is negatively associated with the risk of anxiety. In addition, some studies have shown that people with poor sleep quality have poor psychological stress capacity, which increases the likelihood of anxiety. Several studies have examined the mediating role played by psychological resilience in the association between external influences and mental health. Based on that, we explored the association of sleep quality with anxiety symptoms and the mediating role of psychological resilience in this association. 

5. A potential mechanism by which poor sleep quality affects mental health is that poor sleep quality may affect the neuroendocrine system, particularly leading to activation of the hypothalamic-pituitary-adrenal axis and elevated glucocorticoids, thereby affecting mental health.'

 • This situation is not included in the study data or the introduction part of the article. It is recommended to explain the result for another possible reason or to make a connection with the subject in the introduction.

Thank you for your comments and suggestions. We wanted to explain the results further in terms of biological mechanisms, so we cited that theory for discussion. However, as you say, that potential mechanism cannot be directly suggested by our data. So, we decided to remove that part of the discussion after consideration.

6. ‘Studies also 200 have found a negative correlation between psychological resilience and its various dimensions and anxiety. The possible reason for this is that psychological resilience contributes to positive emotional recovery’

• This statement should be detailed. Positive emotional recovery is often used for depressive processes. Its connection with anxiety should be detailed.

We sincerely appreciate the valuable comments. We have rewritten this part according to suggestions. The details are shown as followed:

In addition, previous studies also have found a negative correlation between psychological resilience and its various dimensions and anxiety [27, 28]. A study investigating the reciprocal relationship of resilience with anxiety symptoms using a three-wave cross-lagged design found that the cross-lagged associations between resilience and anxiety symptoms were stable across time [29]. 

7. ‘Further, some other studies have found an association between sleep quality and psychological resilience. The possible reason for this is that poor sleep quality leads to poor psychological stress capacity.’

• In this study, in which cross-sectional data were analyzed, it is not a direct result that low sleep quality causes psychological stress. However, it can be said that they are related. The explanation should be made with the support of the literature.

Thank you for your comments and suggestions. We have checked the literature and added more references. And we are sorry for your misunderstanding due to our unclear expression. In fact, our study is a longitudinal study, a linear mixed-effects model that took into account time effects and made dynamic measurements, which is different from the cross-sectional study. The details are shown as followed:

The possible reason for this is that poor sleep quality leads to poor psychological stress capacity. A previous meta-analysis demonstrated a clear positive, relationship between sleep and psychological resilience, and put forward that individuals that acquire a sufficient amount and good quality are less likely to suffer from chronic disease, have better hormone regulation, and may also be better cognitively equipped to cope with stressors [31].

8. ‘Therefore, there is a need for prospective studies to explore the association between sleep quality and anxiety.’

• In the study presented in the longitudinal design, but there is only one measurement statistical analysis. It is recommended to elucidate the contribution of the longitudinal pattern to the study.

Thank you for your comments and suggestions. Indeed, in this manuscript, the sleep quality, resilience and anxiety symptoms were all measured twice. We based on this to explore the dynamic association between sleep quality and anxiety symptoms among college students.

-Reviewer 2

1. There are some grammatical errors in the article, and I suggest language editing by the native speaker.

We apologize for the poor language of manuscript. The language of this manuscript has been edited by the native speaker. We have used the Essaystar Language Editing Services (No. P-202207270225uxw) to improve the English of the manuscript.

2. In the last paragraph of the introduction, researchers should include the purpose, hypotheses, and expected possible contributions of the study, not the method and materials of the study.

Thank you for pointing out this problem in the manuscript. We have restructured the last paragraph of the introduction as the following:

Given that, we formulated our hypothesis that poorly sleep quality is associated with anxiety and that resilience plays a mediating role in sleep quality-anxiety association Therefore, we conducted a longitudinal study in order to find the interventions to prevent and treat the anxiety symptoms in a college student population.

3. Researchers should elaborate the introduction based on the hypotheses of the study. Past studies should be utilized, and particularly why college students were evaluated.

Thank you for your comments and suggestions. In the first paragraph of introduction, we have mentioned that prevalence of sleep disorders and mental health were widespread around the world. And, the most of previous studies were based on cross-sectional study to explore the relation between them. So, we proposed that there is a need for prospective studies to explore the association between sleep quality and anxiety in college students. We also have further supplemented the reasons for the focus on college students, and the details are shown as followed: 

Increased caffeine intake, stress and irregular sleep-wake patterns lead to widespread sleep disorders and sleep deprivation in the college population [5]. Previous studies have found that one third of the Chinese college students has problems with sleep quality [6]. In addition, mental health problems (such as anxiety) are widespread in the college population, especially during the novel coronavirus epidemic [7, 8]. Previous studies have found a strong association between the quality of sleep of college students and the onset of their anxiety symptoms [9, 10]. However, most of the studies are based on cross-sectional studies. Therefore, there is a need for prospective studies to explore the association between sleep quality and anxiety in college students.

4. Study participants should be written in more detail. How were the participants reached? Was informed consent obtained? Were any participants excluded from the study? What were the inclusion criteria? 

Thank you for your comments and suggestions. The participants were recruited by a convenience sampling method. All study participants in this study had informed consent. Our inclusion and exclusion criteria were university students who were not hospitalized or treated for infection during the survey period and who did not have a mental disorder prior to the current survey. We have added the information required and the details are shown as followed: 

2.1 study participants

In this study, 127 university students who had given their informed consent, were not hospitalized or treated for infection and during the survey and did not have mental disorders before this survey were recruited for a longitudinal study using a convenience sampling method. The participants study participated in an on-site electronic questionnaire for two repeated measures. The two survey periods of this study were April 5, 2022-April 12, 2022 and May 27, 2022-June 3, 2022, respectively. During this period, the occurrence of the epidemic and the school being under closed management may have affected the sleep status and mental health of some students. This study complied with the Declaration of Helsinki and approved by the Ethics Committee of Fujian medical university (Ethics NO.123 of 2021).

5. How did the researchers determine the sample size? Did they conduct a power analysis?

Thank you for your comments and suggestions. We have conducted a power analysis by PASS software (version 11). Based on our study, we set parameters as the picture below. And the result shows that the sample size of our study has a great power. (Tests for Mean in a Repeated Measures Design, Alpha=0.05, Different to Detect=1, Repeated Measurements=2).

6. Why did the researchers assess generalized anxiety symptoms using a DSM-IV-TR scale instead of DSM-5? 

Thank you for your comments. In this study, we did not use DSM-IV-TR scale to directly assess the generalized anxiety symptoms. Indeed, symptoms were assessed by GAD-7 scale. Although the GAD-7 scale has been developed in accordance with the DSM-IV-TR, it is widely used in the assessment of anxiety symptoms, and has good reliability and validity in Chinese population (PMID: 32642822).

7. The CD-RISC scale description should be reviewed. In the first sentence, it is stated that it consists of 4 dimensions. As far as I know, it has three dimensions: tenacity, strength, and optimism. The dimension names are also written differently, so the authors should check this area. A similar situation is also valid for figure-2. In Figure-2, CD-RISC subdimensions are defined with different names.

Thank you for your correction. We have revised relative expression.

8. Scale names and abbreviations should be expressed in the same terms throughout the manuscript.

Thank you for your comments and suggestions. We checked the manuscript again carefully to ensure consistency of scale names, abbreviations and other terminological expressions.

9. 'it was found that the Cronbach's alpha coefficient of all three scales was greater than 0.6, suggesting that the consistency of all three scales was good.'

It is appropriate to add references for the statement (Lines 133-134).

Thank you for your comments and suggestions. We have supplemented the reference for the statement.

10. Table-3 should be renamed.

Thank you for your comments and suggestions. Table-3 has been renamed.

Comparison of GAD, PSQI and CD-RISC scores between the two surveys.

11. What is the difference between the two separate 'sleeping time' headings in Table-3?

Thank you for your comments and suggestions. To avoid ambiguity, we have rephrased the entry. The details were shown in Table 3.

12. Since a diagnosis of mental illness cannot be made using only the scale, statements such as generalized anxiety disorder, etc., should be avoided, as in Tables 4 and 5. I suggest interpreting the results as anxiety symptoms. Therefore, the authors should rename table-4-5 and figure-2.

Thank you for your comments and suggestions. Table-4-5 and figure-2 has been renamed. The details are shown as followed:

Table 4. Association of sleep quality and psychological resilience with anxiety symptom.

Table 5. Association of sleep quality and psychological resilience with anxiety symptom, stratified by sex and age.

Figure 2. Mediation of association between sleep quality and anxiety symptom by psychological resilience.

13. Models 1 and 2 in Table-4 should be explained in detail in the text.

Thank you for your comments and suggestions. In this manuscript, we mainly focused on the model 2 which explored the association between sleep quality and anxiety symptom after adjusting the potential confounding factors, such as sex, age, BMI and physical exercise rather that model 1 which only examined the association them.

14. Mediation analysis results should be written in detail. Such as whether CD-RISC and its subheadings have a partial or full mediator role. Since anxiety symptoms may also affect sleep quality, why did the researchers not also investigate the relationship between GAD-7 (X) and PSQI (Y) (X→M→Y)? There may be a bidirectional relationship between GAD-7 and PSQI scores.

Thank you for your comments and suggestions. We have detailly supplemented the mediation analysis results. Currently, many studies have found that people with mental disorders sleep worse than people without mental disorders (PMID: 27416139). In addition, the extent to which sleep is causally related to mental health is unclear. However, a meta-analysis showed that improving sleep quality could lead to better mental health (PMID: 34607184 and 30264137). So, in this study, we paid more attention to the relationship between PSQI (X) and GAD-7 (Y) (X→M→Y). We also have supplemented these reasons for our choice in Introduction.

15. In the first paragraph of the discussion (lines 186-190), the authors made inferences about the onset of anxiety symptoms. However, I am not sure that the development of anxiety symptoms can be inferred from the study data. I think the interpretation of increase or decrease would be more appropriate.

Thank you for your comments and suggestions. Indeed, in this manuscript, the sleep quality, resilience and anxiety symptoms were all measured twice. We based on this to explore the dynamic association between sleep quality and anxiety symptoms among college students by mixed linear effects model. So, we think we can reach such a conclusion that poor sleep quality could lead to the onset of anxiety symptoms.

16. The researchers administered the scale to the participants at approximately 1-2 months intervals. During this time, anxiety symptoms and sleep quality may be affected by many variables (students may become infected, hospitalized and receive treatment, etc.), including the epidemic period. Therefore, I do not consider it sufficient to associate the study results only with anxiety and sleep.

Thank you for your comments and suggestions. We apologized for the lack of detail in the description of the study design in our previous manuscript. It was a population-based longitudinal study that assessed university students during the initial and post-closure periods when the campus adopted close management due to the epidemic. In fact, to reduce pathological factors affecting anxiety symptom, infected students were excluded from the study. We have added a description of study participants. The details are shown as followed:

2.1 study participants

In this study, 127 university students who had given their informed consent, were not hospitalized or treated for infection and during the survey and did not have mental disorders before this survey were recruited for a longitudinal study using a convenience sampling method. The participants study participated in an on-site electronic questionnaire for two repeated measures. The two survey periods of this study were April 5, 2022-April 12, 2022 and May 27, 2022-June 3, 2022, respectively. During this period, the occurrence of the epidemic and the school being under closed management may have affected the sleep status and mental health of some students. This study complied with the Declaration of Helsinki and approved by the Ethics Committee of Fujian medical university (Ethics NO.123 of 2021). 

17. References should be organized according to the journal rules.

Thank you for your comments and suggestions. We have reorganized the reference of this manuscript according to journal rules.

---

## [Decision Letter · Decision Letter 1]

15 May 2023

PONE-D-23-03587R1Psychological resilience mediates the association between sleep quality and anxiety symptoms: Base on a longitudinal studyPLOS ONE

Dear Dr. Cai,

Thank you for submitting your manuscript to PLOS ONE. After careful consideration, we feel that it has merit but does not fully meet PLOS ONE’s publication criteria as it currently stands. Therefore, we invite you to submit a revised version of the manuscript that addresses the points raised during the review process.

We look forward to receiving your revised manuscript.

Kind regards,

Mehmet Emin Demirkol

Academic Editor

PLOS ONE

Additional Editor Comments:

Dear Huangjie Cai,

The reviewers completed their assessments. One of the reviewers suggested accepting the manuscript, but the other suggested significant revisions.

Best regards

Reviewers' comments:

Reviewer's Responses to Questions

**Comments to the Author**

1. If the authors have adequately addressed your comments raised in a previous round of review and you feel that this manuscript is now acceptable for publication, you may indicate that here to bypass the “Comments to the Author” section, enter your conflict of interest statement in the “Confidential to Editor” section, and submit your "Accept" recommendation.

Reviewer #1: (No Response)

Reviewer #2: (No Response)

2. Is the manuscript technically sound, and do the data support the conclusions?

Reviewer #1: Partly

Reviewer #2: (No Response)

3. Has the statistical analysis been performed appropriately and rigorously? 

Reviewer #1: I Don't Know

Reviewer #2: (No Response)

4. Have the authors made all data underlying the findings in their manuscript fully available?

Reviewer #1: Yes

Reviewer #2: (No Response)

5. Is the manuscript presented in an intelligible fashion and written in standard English?

Reviewer #1: Yes

Reviewer #2: (No Response)

6. Review Comments to the Author

Reviewer #1: • In the study presented in the longitudinal design, there is only one measurement statistical analysis. It is recommended to elucidate the contribution of the longitudinal pattern to the study. Authors' explanation is that 'sleep quality, resilience and anxiety symptoms were all measured twice. We based on this to explore the dynamic association between sleep quality and anxiety symptoms among college students.' The statistical analysis should be discussed for other measurements.

• The chronological emergence of the factors included in the mediation model should be mentioned. Chronologically, using the previously revealed factor as a mediator does not seem compatible with the model. The authors' explanation is that 'In this study, what we mainly explored is that psychological resilience played the mediating role in this association between sleep quality and anxiety symptoms. Several studies have examined the mediating role played by psychological resilience in the association between external influences and mental health. Based on that, we explored the association of sleep quality with anxiety symptoms and the mediating role of psychological resilience in this association.' Taking references from previous studies is partially accepted. However, its compliance with the basic principles of mediation analysis should be explained within the framework of statistical science.

• In the study, although the participants were observed for a certain period, cross-sectional data was analyzed. Therefore, it does not seem compatible with the longitudinal pattern of the study. The emphasis on the longitudinal pattern should be changed in the title and content of the study.

Reviewer #2: (No Response)

7. PLOS authors have the option to publish the peer review history of their article (what does this mean?). If published, this will include your full peer review and any attached files.

Reviewer #1: No

Reviewer #2: No

---

## [Author Response · Author response to Decision Letter 1]

22 Sep 2023

Jun 27th, 2023

Mehmet Emin Demirkol, MD, PhD 

Editor

PLOS ONE 

Dear Dr. Demirkol, Please find enclosed our revised manuscript entitled “Psychological resilience mediates the association between sleep quality and anxiety symptoms: a repeated measures study in college students (PONE-D-23-03587R1).” We appreciate the issues raised by the reviewers and the editors and are grateful for the opportunity to respond. Below we address the queries raised during the review process.

We believe that the editor and reviewer comments were very helpful and addressing them has strengthened the manuscript, and we hope you agree. If there are any further issues or concerns that we can address, please do not hesitate to contact us. 

Sincerely,

Dr Huangjie Cai,

1 Xuefu North Road, Fuzhou University New District

Email: 165475322@qq.com

Tel: +86 13774515133

Below are the original review comments, and our responses are in blue text under each one.

-Reviewer 1

• In the study presented in the longitudinal design, there is only one measurement statistical analysis. It is recommended to elucidate the contribution of the longitudinal pattern to the study. Authors' explanation is that 'sleep quality, resilience and anxiety symptoms were all measured twice. We based on this to explore the dynamic association between sleep quality and anxiety symptoms among college students.' The statistical analysis should be discussed for other measurements.

Thank you for your comments. We added to the strengths the contribution of repeated measures studies, which may provide an opportunity to explore the triangular causal relationship between sleep quality, anxiety symptoms, and psychological resilience.

• The chronological emergence of the factors included in the mediation model should be mentioned. Chronologically, using the previously revealed factor as a mediator does not seem compatible with the model. The authors' explanation is that 'In this study, what we mainly explored is that psychological resilience played the mediating role in this association between sleep quality and anxiety symptoms. Several studies have examined the mediating role played by psychological resilience in the association between external influences and mental health. Based on that, we explored the association of sleep quality with anxiety symptoms and the mediating role of psychological resilience in this association.' Taking references from previous studies is partially accepted. However, its compliance with the basic principles of mediation analysis should be explained within the framework of statistical science.

• In the study, although the participants were observed for a certain period, cross-sectional data was analyzed. Therefore, it does not seem compatible with the longitudinal pattern of the study. The emphasis on the longitudinal pattern should be changed in the title and content of the study.

Thank you for your comments and suggestions. In this study, we conducted two repeated-measures analyses of the study population, and several similar studies have now used the same study design (PMID: 36743184) and conducted mediation analyses. The mixed linear effects model has now been widely used for the analysis of repeated measures data and its mediation analysis (PMID: 35007609). We have described this in our methodology and cited the relevant literature. In addition, we have changed the emphasis on the longitudinal pattern should be changed in the title and content of the study.

---

## [Decision Letter · Decision Letter 2]

2 Oct 2023

Psychological resilience mediates the association between sleep quality and anxiety symptoms: a repeated measures study in college students

PONE-D-23-03587R2

Dear Dr. Cai,

We’re pleased to inform you that your manuscript has been judged scientifically suitable for publication and will be formally accepted for publication once it meets all outstanding technical requirements.

Kind regards,

Mehmet Emin Demirkol

Academic Editor

PLOS ONE

Additional Editor Comments (optional):

Reviewers' comments:

Reviewer's Responses to Questions

**Comments to the Author**

1. If the authors have adequately addressed your comments raised in a previous round of review and you feel that this manuscript is now acceptable for publication, you may indicate that here to bypass the “Comments to the Author” section, enter your conflict of interest statement in the “Confidential to Editor” section, and submit your "Accept" recommendation.

Reviewer #1: All comments have been addressed

2. Is the manuscript technically sound, and do the data support the conclusions?

Reviewer #1: Yes

3. Has the statistical analysis been performed appropriately and rigorously? 

Reviewer #1: Yes

4. Have the authors made all data underlying the findings in their manuscript fully available?

Reviewer #1: Yes

5. Is the manuscript presented in an intelligible fashion and written in standard English?

Reviewer #1: Yes

6. Review Comments to the Author

Reviewer #1: The authors implemented all recommendations. I have no additional suggestions. The final version of the manuscript was found to comply with the journal rules.

7. PLOS authors have the option to publish the peer review history of their article (what does this mean?). If published, this will include your full peer review and any attached files.

Reviewer #1: No

---

## [Editor Report · Acceptance letter]

8 Oct 2023

PONE-D-23-03587R2 

Psychological resilience mediates the association between sleep quality and anxiety symptoms: a repeated measures study in college students 

Dear Dr. Cai:

I'm pleased to inform you that your manuscript has been deemed suitable for publication in PLOS ONE. Congratulations! Your manuscript is now with our production department. 

Kind regards, 

on behalf of

Dr. Mehmet Emin Demirkol 

Academic Editor

PLOS ONE